# Reversed HILIC Gradient: A Powerful Strategy for On-Line Comprehensive 2D-LC

**DOI:** 10.3390/molecules28093907

**Published:** 2023-05-05

**Authors:** Soraya Chapel, Florent Rouvière, Davy Guillarme, Sabine Heinisch

**Affiliations:** 1Institut Des Sciences Analytiques, Université de Lyon, UMR 5280, CNRS, 5 rue de la Doua, 69100 Villeurbanne, France; soraya.chapel@kuleuven.be (S.C.); florent.rouviere@isa-lyon.fr (F.R.); 2Pharmaceutical Analysis, Department of Pharmaceutical and Pharmacological Sciences, University of Leuven (KU Leuven), Herestraat 49, 3000 Leuven, Belgium; 3School of Pharmaceutical Sciences, University of Geneva, CMU–Rue Michel Servet 1, 1211 Geneva 4, Switzerland; davy.guillarme@unige.ch; 4Institute of Pharmaceutical Sciences of Western Switzerland, University of Geneva, CMU–Rue Michel Servet 1, 1211 Geneva 4, Switzerland

**Keywords:** comprehensive 2D-LC, on-line LC × LC, reversed HILIC, orthogonality, pharmaceuticals, peptides

## Abstract

The aim of the present work is to evaluate the possibilities and limitations of reversed hydrophilic interaction chromatography (revHILIC) mode in liquid chromatography (LC). This chromatographic mode consists of combining a highly polar stationary phase (bare silica) with a gradient varying from very low (1–5%) to high (40%) acetonitrile content (reversed gradient compared to HILIC). The retention behavior of revHILIC was first compared with that of reversed-phase LC (RPLC) and HILIC using representative mixtures of peptides and pharmaceutical compounds. It appears that the achievable selectivity can be ranked in the order RPLC > revHILIC > HILIC with the two different samples. Next, two-dimensional liquid chromatography (2D-LC) conditions were evaluated by combining RPLC, revHILIC, or HILIC with RPLC in an on-line comprehensive (LC × LC) mode. evHILIC × RPLC not only showed impressive performance in terms of peak capacity and sensitivity, but also provided complementary selectivity compared to RPLC × RPLC and HILIC × RPLC. Indeed, both the elution order and the retention time range differ significantly between the three techniques. In conclusion, there is no doubt that revHILIC should be considered as a viable option for 2D-LC analysis of small molecules and also peptides.

## 1. Introduction

Over the past two decades, hydrophilic interaction liquid chromatography (HILIC) has emerged as a powerful technique for the separation of polar and ionizable compounds. In contrast to reversed-phase liquid chromatography (RPLC) which employs a non-polar stationary phase and a polar mobile phase, HILIC uses a polar stationary phase and a less polar mobile phase for the separation, resulting in the preferential retention of polar analytes [1,2,3,4]. Unlike normal-phase liquid chromatography (NPLC), where the mobile phase contains only organic solvents, HILIC mobile phases are usually composed of a mixture of water and aprotic organic solvent (primarily acetonitrile) [2]. Although this topic has been debated for many years, the most widely accepted mechanism for HILIC separation mainly involves hydrophilic partitioning between a water-enriched layer on the surface of this polar stationary phase and the organic-rich mobile phase. Depending on the target compounds, other types of interactions may also be involved, including adsorption, ion exchange, or hydrogen bonding [2,4,5,6,7,8].

Extensive research has been conducted in the past on retention models in HILIC. They revealed U-shaped curves spanning from 0 to 100% water, suggesting the existence of bimodal retention behaviors within this range of mobile phase compositions [1,2,3,4,5]. Typically, the retention of all solutes decreases as the concentration of water increases until about 40% of water, while retention increases beyond this value. For this reason, in gradient elution, HILIC separation is generally accomplished by elevating the concentration of water in the mobile phase from approximately 2–5% to 30–40%, depending on the stationary phase [2,5,9,10,11], while higher composition ranges are rarely explored.

In the early 2010s, a global acetonitrile shortage prompted researchers to investigate solvents other than acetonitrile for the analysis of highly polar ionizable solutes [12]. It was found that using common HILIC polar sorbents, such as bare silica, in combination with a highly aqueous solvent, could provide attractive chromatographic performance for these compounds under isocratic elution conditions. During this decade, several other groups explored this approach and demonstrated its suitability for the analysis of various types of compounds, including amino acids [12], peptides [13], proteins [14], pharmaceuticals [15], synthetic pigments [16], and catecholamines [12,17], among others [12,18,19,20,21,22,23]. This alternative mode was called reversed HILIC or per-aqueous liquid chromatography (PALC) and was initially proposed as a potential replacement for HILIC. Since then, the possibility of using water-rich mobile phases on polar sorbents has been little reported and almost exclusively under isocratic conditions.

HILIC has been proven to be a complementary technique to RPLC for the analysis of polar and ionizable compounds that tend to be poorly retained in RPLC. In recent years, the combination of HILIC and RPLC has thus received tremendous attention for the separation of complex mixtures with a broad range of polarities in two-dimensional liquid chromatography (2D-LC) [24,25]. In fact, this combination is rapidly becoming one of the most widely used in 2D-LC, right after the use of RPLC in both dimensions [24]. In on-line comprehensive two-dimensional liquid chromatography (LC × LC), it has been demonstrated to be a powerful analytical approach for the separation of complex mixtures such as (bio)-pharmaceutical products [26,27,28,29,30,31], natural products [32,33,34,35], food products [36,37,38], and polymeric samples [39,40,41], to cite only a few. Compared to RPLC × RPLC, better orthogonality and larger effective peak capacities have been reported [42,43]. However, employing HILIC in one of the two dimensions can be quite challenging in on-line LC × LC for two main reasons. Firstly, HILIC mode is often unsuitable for injecting the highly aqueous solvents that commonly surround the target sample [2,44]. Secondly, the reversed elution strength of the two mobile phase systems used in HILIC and RPLC (i.e., highly organic in HILIC versus highly aqueous in RPLC) typically leads to poor peak shapes in the second dimension [45]. This phenomenon is commonly known as the solvent strength mismatch problem. To circumvent this, many strategies Fhave been developed and reported over the years [24,45,46,47]. The main techniques employed for addressing this problem involve flow splitting, solvent dilution, trapping, and solvent evaporation between dimensions. These approaches are specifically designed to reduce the volume of the incompatible solvent transferred between the two dimensions and/or substitute it with a more suitable alternative. In this respect, the use of water-rich mobile phases in HILIC could be an attractive alternative.

The aim of the present work is to explore the potential and limitations of using reversed gradients (i.e., with increasing acetonitrile concentrations) instead of normal gradients (i.e., with increasing water concentrations) with polar HILIC stationary phases for the separation of various pharmaceuticals and peptides. First, we deeply investigate the retention behaviors of representative mixtures under reversed HILIC conditions using a bare silica phase. Then, we compare these results, in terms of retention, selectivity, and peak shapes, with RPLC and conventional HILIC. Finally, we present a new approach for on-line comprehensive 2D-LC that uses reversed HILIC in the first dimension (^1^D) and RPLC in the second dimension (^2^D) and compare its performance with RPLC × RPLC and HILIC × RPLC for the analysis in less than 30 min of two complex mixtures of pharmaceuticals and peptides, respectively.

## 2. Results and Discussion

### 2.1. Interest in revHILIC

#### 2.1.1. Retention in HILIC and revHILIC

In the first instance, the logarithm of the retention factor (log k) was plotted against the percentage of water in the mobile phase for seven different peptides, with molecular weights ranging from 555 to 1619 g/mol, and pI ranging from 6 to 12.5. A bare hybrid silica column (Waters BEH HILIC) was used, and the mobile phase consisted of ACN and water in the presence of 10 mM ammonium acetate. Isocratic experiments were performed with water compositions ranging from 5 to 99%. In a few extreme conditions (peptides 6 and 7 at 99% H_2_O; peptide 7 at 5% H_2_O and peptides 3, 4, and 5 at 5% H_2_O), retention was too high (k > 250) and the corresponding data are not included in Figure 1. As highlighted in Figure 1, a significant deviation from linearity was observed over the whole composition range, and three distinct zones can be roughly defined on the retention curves [6]. The boundaries between the three zones are certainly not as precise, since they seem to depend on the compounds. The first region (zone #1) corresponds to the low water content of the mobile phase (between 5 and 40 %). Under these conditions, a layer of water is present on the surface of the stationary phase, leading to the retention of polar analytes, which is mainly based on hydrophilic partitioning. This corresponds to the well-known HILIC mode, where retention decreases with an increasing amount of water in the mobile phase [3,4].

The second region (zone #2) corresponds to the zone where the percentage of water in the mobile phase varies from 40 to 60%. Under these conditions, hydrophilic partitioning in the water layer cannot take place, and retention was very limited for most of the peptides and was probably due exclusively to ionic interactions with the silanols. In this region, k remains approximately low, with minimum k values ranging from 0.6 to 2.7 depending on the peptide, except for peptide 7 (bradykinin, the most basic peptide with a pI of 12.5), which had a minimum k value of about 6. Finally, when the water content exceeded 60% (zone #3), the retention increased again for most peptides, suggesting a change in the retention mechanism. This is the reversed HILIC (revHILIC) region. At high water content, interactions with peptides are probably mainly promoted by the presence of hydrophobic siloxane groups at the surface of the silica material but also possibly by charged silanols through electrostatic interactions [48]. Interestingly, the plots shown in Figure 1 appear to be linear within a limited range of mobile phase compositions, suggesting that the linear solvent strength (LSS) theory [49] can be applied in both HILIC and revHILIC.

#### 2.1.2. Measurement of LSS Parameters in RPLC, HILIC, and revHILIC

Assuming a linear relationship between log k and the mobile phase composition, C, of the strong solvent ($\log k=\log k_{0}-S\times C)$, the two coefficients, S and log *k_0_*, were calculated for all reference compounds (peptides and pharmaceuticals) under RPLC, HILIC, and revHILIC conditions using a procedure that was previously developed and implemented in a commercial modeling software (Osiris 4.2, Euradif, Grenoble, France). The developed strategy is based on two preliminary linear gradients with the same initial composition and two different normalized gradient slopes (with a ratio of at least 3 between the two gradients). A zero search method is then applied to a complex mathematical function derived from the gradient elution differential equation [50]. However, for accurate retention time predictions, it is important to have retention models that are as linear as possible (LSS model), which seems to be only true for a limited range of compositions in HILIC and revHILIC, as shown in Figure 1. Therefore, care should be taken when selecting the conditions of the two preliminary gradients.

In order to assess the linearity of the models in the composition range of interest, experimental retention times were compared to the predicted ones obtained from S and log k0 values calculated according to the procedure described above, for the three different 1D techniques (i.e., RPLC, HILIC, and revHILIC) with the two different samples (peptides and small pharmaceuticals). The experimental retention times were plotted as a function of the predicted retention times in Figure 2.

Two kinds of experimental retention times were interesting: (i) those obtained with a gradient time between the two initial gradient times (interpolation, orange dots in Figure 2), and those obtained with a gradient time shorter or longer than the two initial gradient times (extrapolation, blue and grey dots, respectively, in Figure 2). As shown in Figure 2, the predicted retention times in the three different chromatographic modes were in very good agreement with the experimental ones, with an average deviation often lower than 2% between both, highlighting the validity of the LSS retention models in RPLC, HILIC, and revHILIC within the investigated range of composition. The perfect linearity of the model in revHILIC for peptides was particularly noticeable (Figure 2F).

In the present work, the overall chromatographic performance of the three chromatographic modes was compared, using the average S-values calculated in the six different analytical conditions (described in Section 3.3). The average value of S (S_average_, calculated from all studied compounds) is shown in Figure 3A for the six different studied conditions. In RPLC, S_average_ proved to be quite different for peptides (600 to 1600 g/mol) and for small pharmaceuticals (150 to 600 g/mol), with values of 0.18 and 0.077, respectively. This result is consistent with the fact that a direct correlation exists between the molecular weight of the solute and its S parameter [49]. On the other hand, S_average_ in HILIC was found to be lower, namely 0.10 for peptides and 0.055 for pharmaceuticals. This behavior has recently been reported [51] and could be due to the different interaction mechanisms and the variable interaction energy in HILIC vs. RPLC. Finally, in the case of revHILIC, S_average_ values were quite comparable regardless of the analyzed molecule, namely 0.085 for peptides and 0.073 for pharmaceuticals. Here, again, this behavior was due to the very specific retention mechanism observed in revHILIC, probably mainly based on hydrophobic interactions with non-polar siloxanes on the surface of the stationary phase in the presence of a highly aqueous mobile phase.

Interestingly, Appendix A shows the evolution of S as a function of log k_0_ for the three chromatographic modes, with the peptides and pharmaceutical compounds. In the case of peptides, S increases with retention in RPLC and HILIC modes, while in revHILIC, S decreases with retention. In the latter case, this negative relationship of S vs. log k_0_ has two implications: (i) peak broadening will increase with retention (less compression effect in linear gradient elution mode) and (ii) it might be difficult to elute the most retained compounds before they reach zone #2. To avoid this problem, a shallow gradient should be preferred in revHILIC. For the pharmaceutical compounds, there is no clear trend between S and log k_0_ values in RPLC and HILIC modes. However, similarly to peptides, a clear decrease in S with retention was observed in revHILIC, leading to potential issues on peak broadening and on elution of most retained compounds.

#### 2.1.3. Assessing the Selectivity of revHILIC vs. RPLC and HILIC

The peak capacity, n, in gradient elution can be assessed according to the following relationship:(1)$$n=1+2.3S\Delta C_{e}\times \frac{1}{1+2.3b}\times \frac{\sqrt{N}}{4}$$
where ΔCe is the range of compositions of the strong solvent in the mobile phase at analyte elution covered by the sample, N is the plate count (efficiency), and b is the LSS gradient steepness (b = S × s, with s being the normalized gradient slope)

This equation is very similar to the Purnell equation, which describes the resolution under isocratic conditions. Indeed, these two equations can describe in an independent way the influence of three factors, selectivity, retention, and efficiency, on the quality of the separation either between two consecutive peaks in isocratic elution or between the last and the first peak in gradient elution mode.

In Equation (1), the overall selectivity can be described by the first term of the equation ($S\Delta C_{e}$) [52], while the second term ($\frac{1}{1+2.3b}$) represents the retention and the last term ($\frac{\sqrt{N}}{4}$), is the efficiency.

A simple way to directly compare, for a given sample, the achievable selectivity between different chromatographic conditions is therefore to calculate S × ΔCe, provided that the k_i_ (retention factor under initial gradient composition) is sufficiently large. Such a calculation was carried out for b values close to 0.2 in all conditions.

The calculated ΔCe and S_average_ × ΔCe values are shown in Figure 3B,C, respectively, for the two different samples (peptides and pharmaceuticals) with the three different chromatographic modes (RPLC, HILIC, and revHILIC). For peptides, the highest S_average_ × ΔCe value was obtained in RPLC conditions, followed by revHILIC, and finally HILIC. The superiority of RPLC in terms of selectivity is due to higher S-values in RPLC (in average 0.18) compared to the other chromatographic modes (in average 0.1 and 0.085 in HILIC and revHILIC, respectively) and higher S-values resulting in a larger change in retention for any small change in mobile phase composition. The remarkable differences in the composition ranges covered by the peptides (ΔCe), which varies from 20% in HILIC, 31% in RPLC and up to 41% in revHILIC mode (Figure 3B), also explain the higher S_average_ × ΔCe values (Figure 3C). In terms of selectivity achieved for peptides, the HILIC conditions are clearly the worst (S_average_ × ΔCe equal to 2.0), due to both small S_average_ and above all the limited ΔCe window. revHILIC, therefore, appears as a valuable alternative to HILIC (S_average_ × ΔCe equal to 3.5) for 2D-LC application in combination with RPLC (S_average_ × ΔCe equal to 5.6). Besides the lower selectivity observed in HILIC, there are two additional issues related to the use of HILIC for peptides: (i) some peptides cannot be eluted in HILIC if the b-value is too high, (ii) in the context of proteomics, peptides are generally diluted in water, leading to strong injection effects under HILIC conditions (elution in the breakthrough volume, peak distortion, etc.). It should be noted that the selectivity might change in other conditions (stationary phase, mobile phase pH, additive, etc.) and hence the above values may be different.

For small drugs, RPLC remains the most selective chromatographic mode, followed by revHILIC and HILIC, with S × ΔCe values of 3.8, 1.5, and 0.9, respectively. Whatever the chromatographic mode, selectivity was found to always be superior for peptides vs. small drugs. For small drugs, the ranking was mostly attributed to significant differences between the composition ranges covered by the pharmaceuticals, namely 17% in HILIC, 20% in revHILIC, and 49% in RPLC. It is also important to mention that some drugs were not sufficiently retained under HILIC conditions, which is another significant limitation of this chromatographic mode. On the other hand, the S values were not responsible for the changes in S × ΔCe values under the different chromatographic modes, as they were comparable for all chromatographic modes (0.055, 0.073, and 0.077 in HILIC, revHILIC, and RPLC, respectively). Similarly to the observations made with peptides, our results confirm that revHILIC should be preferred over HILIC when analyzing small drugs. In the case of a multidimensional setup, revHILIC should also be preferentially combined with RPLC, rather than HILIC, to maximize achievable selectivity.

#### 2.1.4. Evaluation of Peak Shapes in revHILIC vs. RPLC and HILIC

Besides selectivity, it is also important to consider the peak shapes obtained under the different chromatographic modes. Figure 4 and Appendix A show the chromatograms obtained in RPLC, HILIC, and revHILIC for the selected peptides and small drugs, respectively, allowing us to evaluate the peak shapes (i.e., width, asymmetry, shouldering). For a reliable comparison, the gradient times were adjusted according to the composition range and mobile phase flow rates to obtain comparable gradient steepness (b close to 0.2) and hence comparable retention, regardless of the chromatographic mode. In Figure 4, the extracted ion chromatograms (EIC) obtained with MS detection are superimposed for more than 30 representative peptides in each chromatographic mode. These chromatograms clearly show the superiority of RPLC in terms of peak distribution, peak shapes, and peak widths.

The excellent RPLC results can be attributed to the specific conditions used in this work, including the use of a CSH C18 stationary phase to limit ionic interactions, combined with elevated temperature (80 °C) to increase solute diffusivity. On the contrary, HILIC gives broad and distorted peaks for the selected peptides. This behavior was mostly attributed to the use of an inappropriate sample diluent (water). Finally, the chromatograms obtained with peptides in revHILIC were quite good even in the presence of an aqueous sample diluent. The observed peaks are sharp, symmetrical, and well-distributed over the chromatogram. For small drugs, similar conclusions can be drawn in terms of performance for the three modes, but it appears that retention in HILIC mode was too limited for a wide range of drugs (elution close to the column dead time in Appendix A). This behavior can be attributed to the fact that several drugs are not sufficiently polar to be retained in HILIC through hydrophilic partitioning, but also to the electrostatic repulsions that can take place between negatively charged drugs and residual silanols.

Even if the chromatograms obtained in revHILIC are not equivalent to those obtained in RPLC, revHILIC seems to be a useful alternative to HILIC for both peptides and pharmaceuticals.

#### 2.1.5. Comparison of Orthogonality between RPLC, HILIC, and revHILIC

In the previous sections, the different modes were compared for one-dimensional liquid chromatography (1D-LC) applications. In this section, the orthogonality between the different modes was assessed to perform 2D-LC analyses.

A wide variety of orthogonality metrics (quantitative measures of the efficacy of separation space utilization) have been proposed in the context of two-dimensional chromatography [53]. In the present case, we have considered one of the simplest and most direct approaches, which consists of plotting the elution composition of the different compounds (peptides or small drugs) in one chromatographic mode against another one and measuring the corresponding correlation coefficient. This approach can be considered as acceptable for systems without data clustering and data outliers, which is our case, as shown in Figure 5 [54].

Unlike the degree of orthogonality proposed earlier [52], the chosen metric does not consider the selectivity, S × ΔCe. Since RPLC was found to be superior to the other two chromatographic modes in terms of selectivity and peak shapes, it was systematically considered as one of the possible dimensions in Figure 4. RPLC combined with HILIC was therefore compared to RPLC combined with revHILIC for the same b-value of 0.2.

Figure 5A shows the elution composition of the peptides in HILIC vs. RPLC. The orthogonality between the two modes was found to be excellent, with no correlation between the elution compositions in RPLC and HILIC (R² equal to 0.02). Interestingly, the peptides were well-distributed over a wide composition range in RPLC (from 2.1 to 32.9% B), whereas the elution window was narrower in HILIC (no elution before 16% B). On the contrary, the orthogonality between RPLC and revHILIC was lower, with an R² value of 0.44. There was indeed a positive correlation between the retention observed in revHILIC and RPLC. This behavior is logical as the retention in both revHILIC and RPLC can be attributed to hydrophobic interactions, either with siloxane groups at the surface of the silica or with C18 alkyl chains. In Figure 5B, it is also clear that retention in revHILIC was quite limited for a wide range of peptides that were eluted with less than 15% B, whereas the most retained peptides in RPLC were eluted at higher mobile phase compositions in revHILIC (up to 30%).

In the case of small molecules, the situation was quite different. Indeed, revHILIC was found to be much more orthogonal to RPLC than HILIC. The correlation coefficients were equal to 0.02 (revHILIC vs. RPLC) and 0.34 (HILIC vs. RPLC). The small drugs were eluted in a wide composition range in RPLC (between 7 and 29% B), whereas the elution range was thinner in revHILIC (3–22% B) and very narrow in HILIC (12–23% B).

In conclusion, the combination of RPLC and HILIC was found to be the most orthogonal in terms of R² for the analysis of peptides, but the combination of revHILIC and RPLC was more interesting when analyzing small drugs.

### 2.2. Applicability of revHILIC in Comprehensive 2D-LC

In the final part of this study, the potential and limitations of utilizing reversed HILIC in on-line comprehensive 2D-LC for the analysis of complex samples of peptides and pharmaceuticals were explored. To accomplish this, six different on-line LC × LC methods, including RPLC × RPLC, HILIC × RPLC, and revHILIC × RPLC, for both peptides and pharmaceuticals were developed and compared, building on the 1D-LC observations made above in Section 2.1. The operating conditions for these two-dimensional (2D) systems were optimized using an in-house calculation tool previously developed in our lab [55]. In brief, the optimization procedure combines predictive calculations and a Pareto-optimality approach to define the best set of conditions for a given analysis time, taking into account both the effective peak capacity and the dilution factor as key performance descriptors. In all cases, the operating conditions were always optimized with the objective of minimizing the dilution factor (thereby maximizing detection sensitivity), while maintaining a sufficiently high peak capacity for the separation. The optimized conditions included the flow rates in both dimensions, the gradient conditions in ^2^D, and the sampling rate of the ^1^D, while certain conditions, such as the ^1^D-gradient time (fixed at 30 min in this study), the mobile phase natures and compositions, the column dimensions, and the column temperatures in both dimensions, were established before optimization. The selection of the latter was heavily based on past research [55,56,57,58,59,60], but this aspect will not be discussed in this work. The optimized conditions were applied to perform on-line RPLC × RPLC, HILIC × RPLC, and revHILIC × RPLC-UV-HRMS analyses of two representative mixtures of peptide or pharmaceutical samples, and the operating conditions for the six developed 2D systems are provided in Table 1 and Table 2, respectively.

Figure 6 shows a comparison of the resulting contour plots obtained for the analyses of the peptide mixture, while Figure 7 depicts the ones obtained for the pharmaceutical mixture. Initially, it can be noted that the three separations show marked differences, highlighting the distinct selectivity of the three LC modes investigated. Furthermore, it is evident that both the size and the peak occupation of the 2D retention space exhibit significant variations between the different LC × LC configurations, as well as between the two samples.

As anticipated and previously noted [59,60,61], the chromatographic peaks in the RPLC × RPLC separations of peptides (Figure 6A) and pharmaceuticals (Figure 7A) are confined to a narrow region and are mainly distributed along an invisible diagonal line that traverses the contour plot. In contrast, the separation space in HILIC × RPLC (Figure 6B and Figure 7B) is more effectively utilized, particularly for peptides, as underlined in previous work [43]. Conversely, in the two revHILIC × RPLC separations (Figure 6C and Figure 7C), the chromatographic peaks appear to be concentrated in the bottom right-hand side of the 2D space, while the upper left corner is empty.

The 2D retention space coverage obtained for these six separations was estimated using the Stoll–Gilar bin–box method [54,62,63]. In short, this method entails partitioning the 2D separation space into a grid containing n bins of equal size before counting the number of bins that contain at least one chromatographic peak. The coverage of the retention space is subsequently calculated by dividing the number of bins occupied by the total number of bins in the 2D space. In the current study, the total number of bins was not chosen arbitrarily but rather determined based on the number of analytes present in the mixture, as recommended by Gilar et al. [42,54] (i.e., n ~ 67 for the pharmaceutical mixture and n ~ 196 for the peptide mixture). A visual illustration of the determination of the retention space coverage for the six separations can be found in Appendix A. For peptides, the coverages were estimated to be 0.51, 0.83, and 0.43 for the RPLC × RPLC, HILIC × RPLC, and revHILIC × RPLC, respectively. For pharmaceuticals, they were estimated to be 0.58, 0.56, and 0.66, respectively, which is in good agreement with the previous observations made in Section 2.1.5 based on orthogonality diagrams.

All the performance metrics in terms of separation power calculated for these six separations are given in Table 3 and Table 4. They include the under-sampling correction factors (α), the retention space coverages (γ), the ranges of retention times in both dimensions (^1^Δt and ^2^Δt), the average peak widths at 4σ in both dimensions (^1^w_4σ_ and ^2^w_4σ_), and the effective peak capacities of the 2D separations (n_2D,effective_). All the theory supporting these calculations has been previously described in detail [43,56], and the equations used in this study can be found in the table’s footnotes. It should be noted that the peak widths in the second dimension were all determined from HRMS data (extracted ion chromatograms) and not from UV data, due to the chromatogram’s complexity. For this reason, the effective peak capacity values given in this work are expected to be much lower than in reality. The peak widths measured in HRMS are indeed larger than in UV, due to additional extra-column dispersion [60].

As expected, HILIC × RPLC gave the highest peak capacity values for peptides (i.e., 970), while RPLC × RPLC and revHILIC × RPLC gave comparable results (i.e., 571 and 560, respectively). This can be explained by the larger surface coverage and very sharp peaks obtained under total breakthrough conditions [43,65,66] in ^2^D for HILIC × RPLC. On the other hand, the effective peak capacities were the lowest in HILIC × RPLC for pharmaceuticals (i.e., 341) as a result of the small surface coverage and large peak widths in ^2^D, due to injection solvent effects arising from the severe solvent-strength mismatch between dimensions. Despite a larger surface coverage, the effective peak capacity achieved in revHILIC × RPLC for pharmaceuticals (i.e., 593) was lower than the one achieved in RPLC × RPLC (i.e., 886). This is due to a much lower separation space in ^1^D (i.e., 23 min in revHILIC vs. 30 min in RPLC) and larger peak widths in ^2^D (i.e., 0.34 min vs. 0.29 min). Again, these results are consistent with the observations made in Section 2.1.

Figure 8 and Figure 9 show a comparison between 2D-chromatograms overlays and 3D plots obtained in HILIC × RPLC vs. revHILIC × RPLC for the peptide mixture and the pharmaceutical mixture, respectively.

As can be seen, for both samples, the method sensitivity was much higher in revHILIC × RPLC (Figure 8A,C and Figure 9A,C) compared to HILIC × RPLC (Figure 8B,D and Figure 9B,D). It should be noted that the intense peaks observed in the light blue fraction in Figure 9A were not considered for the comparison, as this fraction corresponds to the ^1^D-breakthrough peak. In this work, the peak intensities were on average 6-fold higher in revHILIC × RPLC compared to HILIC × RPLC for peptides, and more than 8-fold higher for pharmaceuticals. There are two main reasons for these differences. Firstly, in ^1^D-revHILIC, larger volumes of aqueous samples could be injected without encountering issues with peak shape, in contrast to ^1^D-HILIC, where poor peak shapes and breakthrough occurred, despite lower injected volumes. A good example of this can be found in Figure 7B and Figure 9B, in which we can see the occurrence of breakthrough phenomena in both dimensions. Secondly, the fairly good compatibility of solvents between dimensions in revHILIC × RPLC led to relatively good peak shapes in ^2^D, unlike in HILIC × RPLC, where poor peak shapes and breakthrough were observed. Those results highlight undoubtedly the great potential of (1) using revHILIC instead of HILIC for the analysis of aqueous samples in the first dimension and (2) employing revHILIC instead of HILIC prior to RPLC to prevent solvent strength mismatch between dimensions.

revHILIC × RPLC not only delivered impressive performance in terms of peak capacity and sensitivity but also provided complementary selectivity when compared to RPLC × RPLC and HILIC × RPLC. Figure 10 shows a comparison of 2D contour plots obtained for eight selected extracted ion chromatograms in RPLC × RPLC (Figure 10A,D), HILIC × RPLC (Figure 10B,E), and revHILIC × RPLC (Figure 10C,F) for both the peptide sample (Figure 10A–C) and the pharmaceutical sample (Figure 10D–F).

The first observation that can be made is that, for the two samples, both the elution order and the retention time range differ significantly between the three techniques. For instance, in the pharmaceutical sample, the elution order is reversed when comparing peaks #7 and #8 in RPLC × RPLC (Figure 10D) versus HILIC × RPLC (Figure 10E) and revHILIC × RPLC (Figure 10F). In fact, while the two peaks are coeluted and eluted between 20 and 25 min in RPLC × RPLC and HILIC × RPLC, they are well-resolved and eluted between 5 and 10 min in revHILIC × RPLC. Similarly, the six highlighted peptides elute within a narrow range in ^1^D and exhibit several coelutions in RPLC × RPLC (Figure 10A) and HILIC × RPLC (Figure 10B) but are conversely nicely spread and separated in revHILIC × RPLC (Figure 10C). It is also noteworthy that in revHILIC × RPLC, the ^1^D-peak widths do not seem to increase significantly with increasing retention times, contrary to what was suggested in Section 2.1. A better understanding of the aforementioned statement can be achieved by referring to Figure 10C, in which the six highlighted peptides clearly exhibit constant peak widths in both dimensions. This once again proves the good chromatographic performance of revHILIC in gradient elution conditions and its suitability for the analysis of complex pharmaceutical and peptide mixtures in on-line LC × LC.

## 3. Materials and Methods

### 3.1. Chemicals and Reagents

Ultra-pure water (Milli-Q^®^) was produced in the laboratory using an Elga Purelab Classic UV purification system from Veolia Water STI (Décines-Charpieu, France). Methanol (MeOH, LC-MS grade), acetonitrile (ACN, LC-MS grade), formic acid (FA, LC-MS grade), and ammonium acetate (AA, analytical reagent grade) were purchased from Fisher Scientific (Illkirch, France). Seven reference standards of peptides, including leucine encephalin, bombesin, [arg8]-Vasopressin, [ile]-angiotensin, bradykinin fragment 1–5, substance P, and bradykinin, were purchased from Merck (Molsheim, France). A detailed list of their physical properties can be found in Appendix A. DL-1,4-dithiothreitol (DTT, 99%) and iodoacetamide (98%), used in the enzymatic digestion of the six model proteins as reducing and alkylated agents, respectively, were purchased from Acros Organics (Geel, Belgium). Trypsin, human serum albumin (HSA), bovine serum albumin (BSA), β-casein, myoglobin, lysozyme, and cytochrome C were all purchased from Sigma-Aldrich (Steinheim, Germany). The sixty-seven reference standards of pharmaceuticals mentioned in this study were purchased from Sigma-Aldrich. A detailed list can be found in Appendix A.

### 3.2. Sample Preparation

The representative peptide sample analyzed in this work was obtained by tryptic digestion of six proteins (HSA, BSA, β-casein, myoglobin, lysozyme, and cytochrome C) using a protocol described in detail in another paper [1,2]. For all experiments, the supernatant of the reaction was injected in the column without dilution after filtration on a 0.22-µm PVDF (polyvinylidene fluoride) membrane. For the standard peptide mixture used to follow the HILIC retention modes, stock solutions of each reference standard were prepared in pure water at a concentration of 500 mg/L for bombesin, [arg8]-vasopressin, [ile]- angiotensin, and substance P, 1000 mg/L for bradykinin fragment 1–5, 2500 for bradykinin, and 5000 mg/L for leucine enkephalin. The final mixture was obtained by mixing appropriate volumes of each stock solution with water and ACN to obtain a 16 µg/mL concentration in 50:50 ACN/water (*v*/*v*%).

For the representative pharmaceutical sample, stock solutions of sixty-seven standard drugs (cf. Appendix A) were prepared in pure MeOH at a concentration of 2 mg/mL. The final sample was made by mixing appropriate volumes of the stock solutions with water to obtain a 40 µg/mL concentration in 11:89 MeOH/water (*v*/*v*%).

### 3.3. Instrumentation

The 1D-LC and 2D-LC analytical measurements described in this work were all carried out on a 1290 Infinity II series 2D-LC system from Agilent Technologies (Waldbronn, Germany). For the 1D-LC measurements, only the first dimension of this system was used. The system consisted of two 1290 high-pressure binary pumps, a 1290 auto-sampler with a flow-through needle injector and a 20 µL storage loop, two column oven with low-dispersion preheaters, and two diode-array ultra-violet (UV) absorbance detectors (DAD) with 0.6 µL flow-cells. UV data were acquired at a rate of 5 Hz and 80 Hz in the first and second dimensions for the 2D-LC experiments, respectively, and 80 Hz for the 1D-LC experiments. The first and second dimensions were connected using a 2-position/4-port duo valve configured in back-flush (also called counter-current or first-in–first-out) mode and mounted with a set of two identical storage loops, whose volume was adapted to the volume of the transferred fractions and depended on the 2D-conditions (cf. Table 1). A pressure release kit (PRK) was installed between the ^1^D-outlet and the 2D-LC valve inlet to protect the ^1^D-detector flow cell from the pressure pulses arising from the successive valve switching. The dwell volumes in the first and second dimensions were, respectively, estimated to be 170 µL and 80 µL (loop volume excluded), while the extra-column volumes were estimated to be 22 µL and 8.5 µL. Agilent OpenLab CDS Chemstation edition (version 2.3.0468) software with Agilent 1290 Infinity 2D-LC add-on (version A.01.04 [025]) was used to operate the 2D-LC system, control the 2D-LC valve, and acquire both the 1D-LC-UV and 2D-LC-UV data.

The chromatographic instrument was coupled to a quadrupole-time-of-flight (Q-TOF) high-resolution mass spectrometer (G6560B series) with a Jet Stream electrospray ionization (ESI) source from the same provider. A homemade flow splitter, consisting of a zero-dead volume tee-piece and appropriate PEEK tubing dimensions, was used to split the effluent from the second dimension (2:1) between the 2D-DAD and the Q-TOF instrument. Agilent Mass Hunter software (version 7.1.7133) was used to control the Q-TOF instrument and acquire both the 1D-LC-HRMS and 2D-LC-HRMS data. The latter were acquired in 2 GHz extended dynamic mode with a scan range of from 100 Da to 3200 Da in ESI positive (+) ion mode. Mass spectra were acquired at a scan rate of 20 spectra/s. The drying gas was set to a temperature of 300 °C and a flow rate of 11 L/min, while the sheath gas was set to a temperature of 350 °C and a flow rate of 11 L/min. The nebulizer gas pressure was set at 40 psi. The capillary, nozzle, fragmentor, and Oct 1 RV voltages were set at 3500, 300, 150, and 750 V, respectively.

Data were analyzed, processed, and visualized using Microsoft Excel, Agilent MassHunter qualitative analysis software (version B.08.00), and an in-house script developed on MATLAB.

### 3.4. Analytical Methods

#### 3.4.1. 1D-LC Methods

HILIC retention curves were obtained by injecting a standard peptide mixture of seven compounds on an Acquity BEH HILIC column (50 mm × 2.1 mm; 1.7 µm) from Waters Technologies (Milford, MA, USA). The column temperature and the flow rate were set at 30 °C and 0.5 mL/min, respectively, and the injected volume was 1 µL. Several isocratic mobile phase elution runs were performed from 5:95 (*v*/*v*%) A/B to 1:99 (*v*/*v*%) A/B with 10 mM AA in water used as solvent A and ACN as solvent B.

For the 1D-RPLC experiments conducted with the representative peptide and pharmaceutical samples, the separations were performed on an Acquity CSH C18 column (50 mm × 2.1 mm; 1.7 µm) from Waters. The column temperature and flow rates were 80 °C and 2.1 mL/min, respectively. The mobile phase was composed of water with 0.1% FA as solvent A and ACN with 0.1% FA as solvent B. Gradient runs were carried out from 1 to 99% of solvent B in various gradient times including 0.4, 0.6, 0.8, 1.6, 2.4, 4.0, and 8.0 min.

For the 1D-HILIC experiments, an Acquity BEH HILIC column (50 mm × 2.1 mm; 1.7 µm) from Waters was used, with a column temperature and a flow rate set at 30 °C and 0.5 mL/min, respectively. The mobile phase was composed of water with 10 mM AA as solvent A and acetonitrile as solvent B. Gradient runs were carried out from 2 to 42% of solvent A in similar gradient times as for the 1D-RPLC experiments.

For the 1D-revHILIC experiments, the experimental conditions were the same as for 1D-HILIC, except that the gradient runs were carried out from 1 to 51% of solvent B in 0.8, 1.6, 3.2, 4.8, 8.0, and 16 min, respectively.

For all six experiments, the volumes injected with each gradient time were set according to the calculated peak variance, as described in previous work [59,60] and were equal to 1.6, 2.0, 2.9, 3.7, 5.4, and 9.6 µL.

#### 3.4.2. On-Line LC × LC Methods

The chromatographic conditions used in the first and second dimensions in the six on-line LC × LC methods developed in this work for the analysis of the representative peptide and pharmaceutical samples are summarized in Table 1 and Table 2, respectively.

## 4. Conclusions

This work highlights the potential of revHILIC as an alternative strategy for the analysis of small drugs and peptides. This chromatographic mode was systematically compared with RPLC and HILIC modes.

In the first part of this work, revHILIC was investigated on the basis of solute retention behavior. The retention models in revHILIC appear to be linear in a large range of compositions, but this was sufficient to accurately measure the LSS parameters (S and log k_0_). For peptides, S values were found to be slightly lower than those in HILIC and significantly lower than those in RPLC but comparable in the three modes for small molecules. The achievable selectivity in gradient elution was evaluated based on the product of S values and ΔC_e_. Irrespective of the two samples analyzed, revHILIC always provides higher selectivity than HILIC.

In the second part of this work, several comprehensive 2D-LC analyses were performed, in less than 30 min, using either RPLC, HILIC, or revHILIC in the first dimension, combined with RPLC in the second dimension. Taking into account all the metrics for the fon-line LC × LC, namely undersampling correction factor, retention space coverage, retention range in both dimensions, and average peak widths, the effective peak capacity could be easily estimated. HILIC × RPLC gave the highest peak capacity for peptides, followed by RPLC × RPLC and revHILIC × RPLC, which gave comparable results. On the other hand, HILIC × RPLC gave the lowest effective peak capacity for small drugs. revHILIC × RPLC and RPLC × RPLC offer an increase in peak capacity of around 70% and 160%, respectively. Sensitivity was also improved by 6 to 8 times for revHILIC × RPLC compared to HILIC × RPLC. Last but not least, revHILIC × RPLC also provides complementary selectivity when compared to RPLC × RPLC and HILIC × RPLC (elution order and retention time range are very different between the three techniques). Finally, it is clear that revHILIC under gradient conditions is an interesting strategy for the analysis of small drugs and peptides and should be considered more and more in the future in on-line LC × LC.

## Figures and Tables

**Figure 1 molecules-28-03907-f001:**
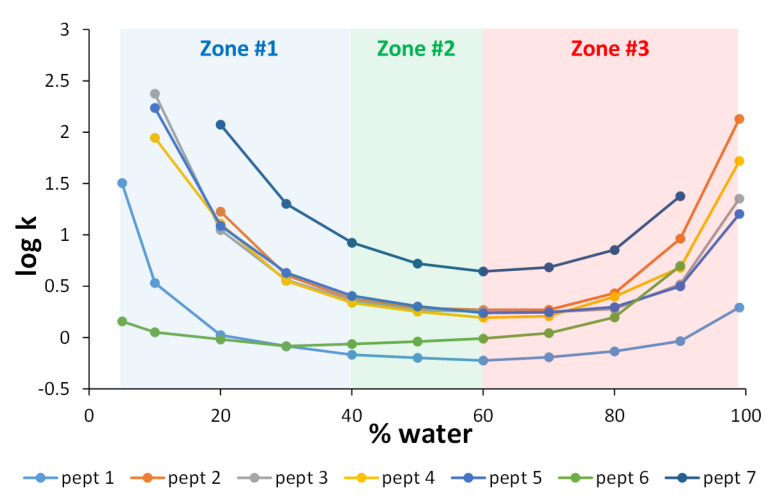
Log k vs. % water for 7 different peptides on the Waters BEH HILIC column at 30 °C and 0.5 mL/min. The mobile phase is composed of ACN and water in the presence of 10 mM ammonium acetate. Various isocratic experiments were performed with compositions of water ranging from 5 to 99%. The three different zones highlighted in the figure are described in Section 2.1.1.

**Figure 2 molecules-28-03907-f002:**
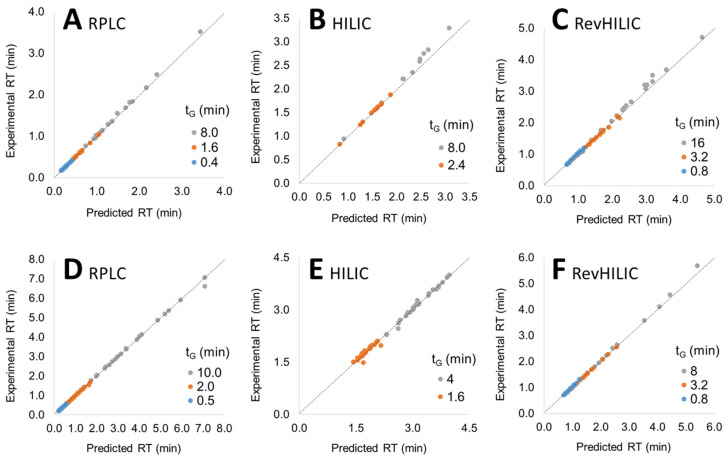
Experimental vs. predicted retention times for various gradient times (t_G_) with pharmaceuticals (**A**–**C**) and peptides (**D**–**F**) in RPLC (**A**,**D**), HILIC (**B**,**E**) and revHILIC (**C**,**F**). The color coding is explained in the text.

**Figure 3 molecules-28-03907-f003:**
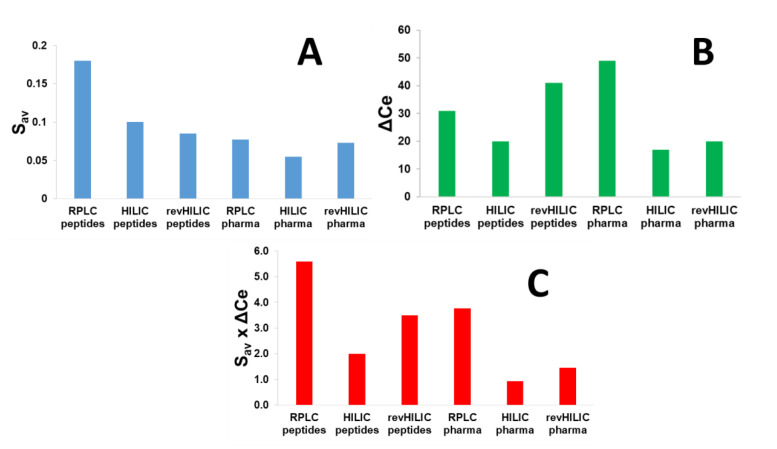
S_average_, ΔCe, and S_average_ × ΔCe values (cf. Equation (1)) for six different chromatographic conditions, including RPLC, HILIC, and revHILIC applied to the separation of a wide range of peptides and small pharmaceutical compounds. (**A**) S_average_ values for the different samples in RPLC and HILIC, (**B**) ΔC_e_ values for the different samples in RPLC and HILIC, (**C**) S_average_ × ΔC_e_ values for the different samples in RPLC and HILIC.

**Figure 4 molecules-28-03907-f004:**
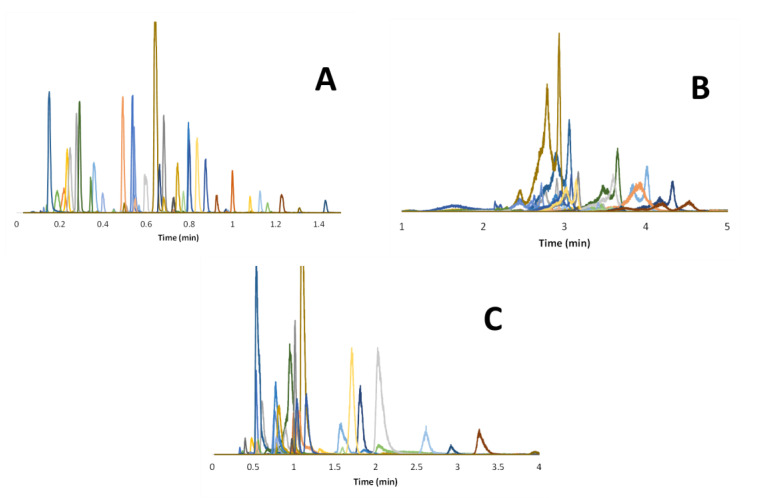
Overlay of numerous representative EICs of peptides in the three different modes. (**A**) RPLC analysis with a 1–99% B gradient in 4 min (b_av_ = 0.25); mobile phase composed of water with 0.1% FA as solvent A and ACN with 0.1% FA as solvent B. (**B**) HILIC analysis with a 2–42% A gradient in 4 min (b_av_ = 0.24). (**C**) revHILIC analysis with a 1–51% B gradient in 4.8 min (b_av_ = 0.21); mobile phase composed of water with 10 mM AA as solvent A and acetonitrile as solvent B. A given color corresponds to a given EIC.

**Figure 5 molecules-28-03907-f005:**
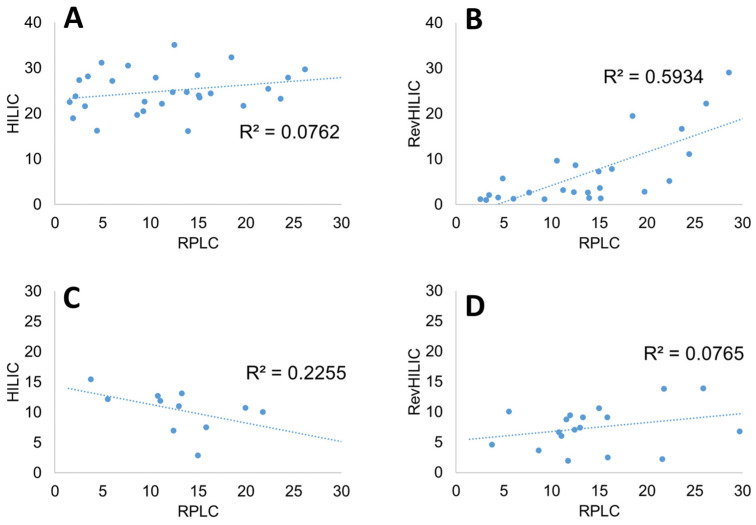
Orthogonality plots expressed as the composition Ce (%) of strong solvent at analyte elution for various combinations of chromatographic dimensions. (**A**) HILIC and RPLC for peptides, (**B**) revHILIC and RPLC for peptides, (**C**) HILIC and RPLC for pharmaceutical compounds, and (**D**) revHILIC and RPLC for pharmaceutical compounds.

**Figure 6 molecules-28-03907-f006:**
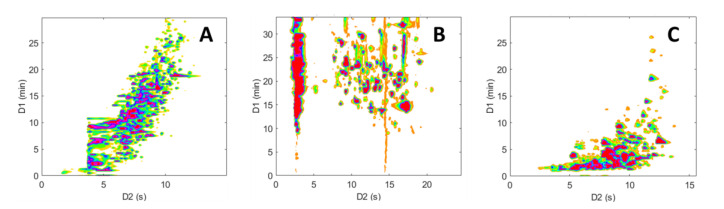
Two-dimensional contour plots (HRMS detection, base peak chromatogram BPC) obtained for the on-line LC × LC separations of the representative peptide sample. (**A**) RPLC × RPLC, (**B**) HILIC × RPLC, and (**C**) revHILIC × RPLC. Chromatographic conditions are given in Table 1.

**Figure 7 molecules-28-03907-f007:**
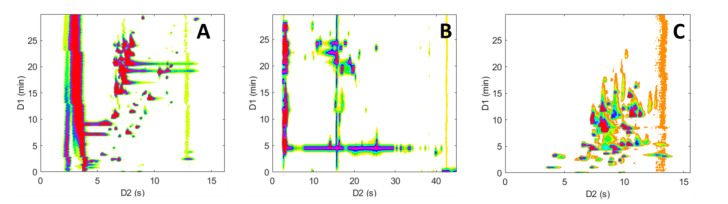
Two dimensional contour plots (HRMS detection, base peak chromatogram BPC) obtained for the on-line LC × LC separations of the representative pharmaceutical sample. (**A**) RPLC × RPLC, (**B**) HILIC × RPLC, and (**C**) revHILIC × RPLC. Chromatographic conditions are given in Table 1.

**Figure 8 molecules-28-03907-f008:**
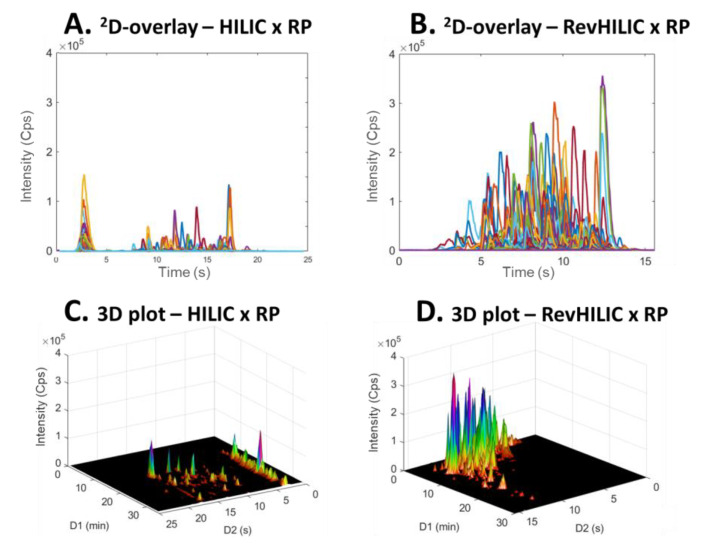
^2^D-chromatogram overlays (**A**,**B**) and three dimensional plots (**C**,**D**) obtained in on-line LC × LC with the representative peptide sample. (**A**,**C**) HILIC × RPLC and (**B**,**D**) revHILIC × RPLC. HRMS detection, base peak chromatogram.

**Figure 9 molecules-28-03907-f009:**
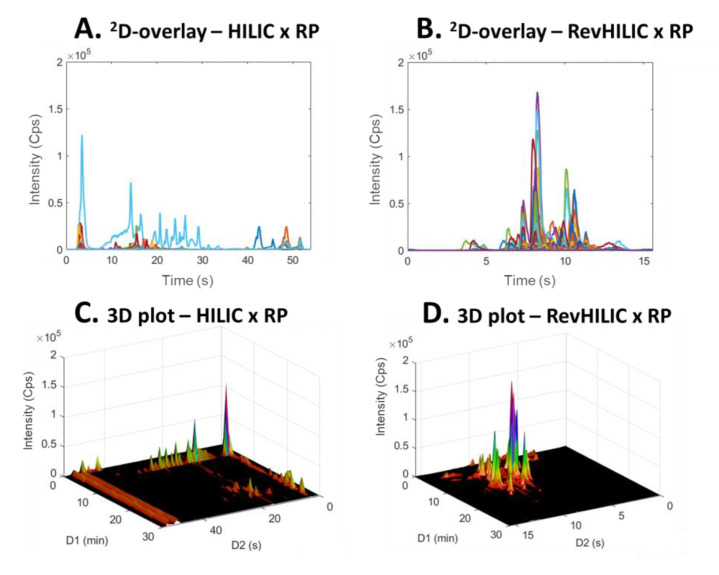
^2^D-chromatogram overlays (**A**,**B**) and 3D three dimensional plots (**C**,**D**) obtained in on-line LC × LC with the representative pharmaceutical sample. (**A**,**C**) HILIC × RPLC and (**B**,**D**) revHILIC × RPLC. HRMS detection, base peak chromatogram.

**Figure 10 molecules-28-03907-f010:**
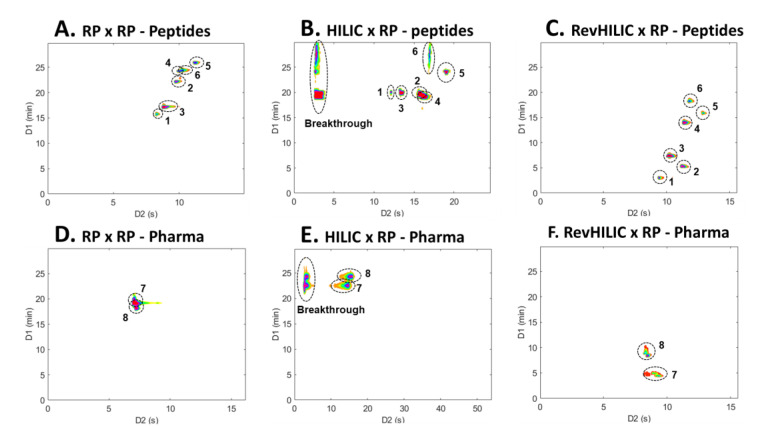
Two dimensional contour plots (HRMS detection) of selected extracted ion chromatograms (EICs) in (**A**,**D**) RPLC × RPLC, (**B**,**E**) HILIC × RPLC, and (**C**,**F**) revHILIC × RPLC for the separation of the representative (**A**–**C**) peptide and (**D**–**F**) pharmaceutical samples. Extracted ion chromatograms (EICs): 1: 1636.8220, 2: 908.8799, 3: 1383.7090, 4: 1112.5860, 5: 1548.2860, 6: 1585.7750, 7: 260.1651, 8: 272.2015.

**Table 1 molecules-28-03907-t001:** Experimental conditions used in on-line RPLC × RPLC, HILIC × RPLC, and revHILIC × RPLC for the analysis of the representative peptide sample.

	RPLC × RPLC	HILIC × RPLC	revHILIC × RPLC
**First dimension**			
Injection volume	5.8 µL	1.8 µL	3.1 µL
Stationary phase	Acquity CSH C18	Viridis BEH HILIC	Viridis BEH HILIC
Column geometry	30 × 2.1 mm; 1.7 µm	50 × 2.1 mm; 1.7 µm	50 × 2.1 mm; 1.7 µm
Temperature	30 °C	30 °C	30 °C
Mobile phase A	Water + 10 mM AA	Water + 10 mM AA	Water + 10 mM AA
Mobile phase B	ACN	ACN	ACN
Flow rate	0.20 mL/min	0.05 mL/min	0.14 mL/min
Gradient	1–36% B in 30 min	10–52% A in 30 min	1–40% B in 30 min
**Modulation**			
Loop volume	60 µL	60 µL	80 µL
Sampling time	0.25 min	0.41 min	0.26 min
**Second dimension**			
Stationary phase	Acquity CSH C18	Acquity CSH C18	Acquity CSH C18
Column geometry	30 × 2.1 mm; 1.7 µm	30 × 2.1 mm; 1.7 µm	30 × 2.1 mm; 1.7 µm
Temperature	80 °C	80 °C	80 °C
Mobile phase A	Water + 0.1% FA	Water + 0.1% FA	Water + 0.1% FA
Mobile phase B	ACN + 0.1 % FA	ACN + 0.1 % FA	ACN + 0.1 % FA
Flow rate	2.6 mL/min	2.6 mL/min	2.6 mL/min
Gradient	1–45% B in 0.13 min	1–45% B in 0.33 min	1–45% B in 0.15 min

**Table 2 molecules-28-03907-t002:** Experimental conditions used in on-line RPLC × RPLC, HILIC × RPLC, and revHILIC × RPLC for the analysis of the representative pharmaceutical sample.

	RPLC × RPLC	HILIC × RPLC	revHILIC × RPLC
**First dimension**			
Injection volume	8.3 µL	4.5 µL	5.5 µL
Stationary phase	Acquity CSH FP	Viridis BEH HILIC	Viridis BEH HILIC
Column geometry	30 × 2.1 mm; 1.7 µm	50 × 2.1 mm; 1.7 µm	50 × 2.1 mm; 1.7 µm
Temperature	30 °C	30 °C	30 °C
Mobile phase A	Water + 10 mM AA	Water + 10 mM AA	Water + 10 mM AA
Mobile phase B	MeOH	ACN	ACN
Flow rate	0.17 mL/min	0.04 mL/min	0.17 mL/min
Gradient	1–79% B in 30 min	2–30% A in 30 min	1–31% B in 30 min
**Modulation**			
Loop volume	60 µL	60 µL	80 µL
Sampling time	0.27 min	0.9 min	0.26 min
**Second dimension**			
Stationary phase	Acquity CSH C18	Acquity CSH C18	Acquity CSH C18
Column geometry	30 × 2.1 mm; 1.7 µm	30 × 2.1 mm; 1.7 µm	30 × 2.1 mm; 1.7 µm
Temperature	80 °C	80 °C	80 °C
Mobile phase A	Water + 0.1% FA	Water + 0.1% FA	Water + 0.1% FA
Mobile phase B	ACN + 0.1 % FA	ACN + 0.1 % FA	ACN + 0.1 % FA
Flow rate	2.6 mL/min	2.6 mL/min	2.6 mL/min
Gradient	1–99% B in 0.15 min	1–99% B in 0.78 min	1–99% B in 0.14 min

**Table 3 molecules-28-03907-t003:** Performance metrics for the on-line LC × LC separations shown in Figure 6 including effective peak capacities (n_2D,effec_), under-sampling correction factors (α), retention space coverages (γ), ranges of retention times in both dimensions (^1^Δt and ^2^Δt), and average peak widths at 4σ in both dimensions (^1^w_4σ_ and ^2^w4_σ_).

	α^a^	γ^b^	^1^Δt (min)	^2^Δt (s)	^1^w_4σ_ (min)	^2^w_4σ_ (s)	n_2D,effec_
RPLC × RPLC	0.25	0.51	30	10.6	0.07	0.40	571
HILIC × RPLC	0.38	0.83	25.4	15.9	0.18	0.26	971
revHILIC × RPLC	0.34	0.43	26.8	11.8	0.10	0.31	560

^a^ Calculated using α=11+0.216τ2 according to [64], with τ, the sampling rate. ^b^ Estimated using the Gilar–Stoll bin box method [54,62,63]. ^c^ Calculated using $n_{2D,effec}=\alpha \times \gamma \times \left(1+\frac{{\;}^{1}t}{{\;}^{1}w_{4\sigma }}\right)\times \left(1+\frac{{\;}^{2}t}{{\;}^{2}w_{4\sigma }}\right)$.

**Table 4 molecules-28-03907-t004:** Performance metrics for the on-line LC × LC separations shown in Figure 7, including effective peak capacities (n_2D,effect_), under-sampling correction factors (α), retention space coverages (γ), ranges of retention times in both dimensions (^1^Δt and ^2^Δt), and average peak widths at 4σ in both dimensions (^1^w_4σ_ and ^2^w_4σ_).

	α	γ	^1^Δt (min)	^2^Δt (s)	^1^w_4σ_ (min)	^2^w_4σ_ (s)	n_2D,effec_
RPLC × RPLC	0.35	0.58	30	12.5	0.11	0.29	886
HILIC × RPLC	0.48	0.56	27	40	0.53	0.60	341
revHILIC× RPLC	0.47	0.66	23.5	10.8	0.15	0.34	593

## Data Availability

The data presented in this study are available on request from the corresponding author.

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
