# Peer review of "Reversed HILIC Gradient: A Powerful Strategy for On-Line Comprehensive 2D-LC"

_molecules, 2023, doi:10.3390/molecules28093907_

Round 1

Reviewer 1 Report

In this manuscript, the authors evaluated the ability of revHILIC to analyze small drugs and peptides and demonstrated the advantages of revHILIC, better sensitivity and higher selectivity, in 1D- and 2D-chromatographic analyses, which provides novel insight into analytical strategies for biomolecules. Below are some comments about the manuscript.

Major issues:

Page 3, first paragraph: What does the “log k” represent here? Retention factor? The authors should explain the meaning of “log k” and its values in the manuscript.

Page 4, first paragraph: Authors should introduce the meaning of “S”, “log K0” and their values in the main text.

Figure 5, first paragraph: How was the average S values calculated? Were they based on the results of different runs or different analytes?

Figure 1: Based on the graph, the log k keeps constant and low between 40% and 80% of water. Does it mean that zone #2 should be from 40% to 80% of water. Detailed information of the peptides, such as sequence, protein source, should be provided for readers’ reference.

Figure 2: What do the values shown in the legend of each panel represent?

Figure 6: Why the gradient of the second-dimension chromatography is much quicker than that of first dimension?

Even if the 2D retention space coverage of revHILIC x RPLC is the smallest compared with RPLC x RPLC and HILIC x RPLC (Figure 7), its resolving power is much stronger than the other two for both peptide and pharmaceutical small molecules (Figure 8, 9, 10). Rev-HILIC x RPLC can separate the analytes within a shorter time and get the mixed analytes well resolved, suggesting its advantages in the analysis of complicated analytes. Have the authors tried with some real protein or pre-clinical samples to see if the combination of Rev-HILIC x RPLC helps to improve the coverage of analytes identification?

Mino issues:

Page 5, line 185: Please change “apolar” to “a polar”.

Figure 2: Typically, experimental retention times are plotted as a function of predicted retention times. For the axis titles, I would recommend using “RT” or “tr” where “r” is subscripted.

Figure 4: Please change “2 – 42% B” to “2-42% A” and include the components of A and B in the legend of Figure 4.

Figure 8 and 9: Please include “small drug” or “peptides” in the legends of the figures to indicate the analytes in the figures.

In the manuscript, the number in “1D” or “2D” sometimes was superscripted. Please use “1D” or “2D” instead of other forms in the manuscript to maintain uniformity.

Reviewer 2 Report

This work describes the implementation of the reversed HILIC or per-aqueous LC mode as a first dimension in comprehensive 2D-LC, whereby the reversed phase mode is used in the second dimension. The approach is interesting as it overcomes to some extent the limitations of the HILICxRPLC and of the RPLCxRPLC combinations. The first is hindered by a typical rather low flow rate in the first HILIC dimension. The second inevitably entails less orthogonality. Because the reversed HILIC mode is not well developed yet, this work illustrates the potential thereof as an orthogonal more compatible technique with RPLC in LCxLC. The work is well performed and well written. Only minor comments appear necessary to be addressed.

Line 102: the title “2.1” is not very clear or informative

Line 124. The authors say that when the water content exceed 80% there is a change in mechanism and we are in zone #3. However, in figure 1 zone #3 starts already at 60% of water content, even though until 70% log k looks still constant. It is not clear where actually zone #3 should start? The text and work should be in line with the graph.

Line 129. It is suggested that the relation log k vs. % water is linear in limited ranges, while looking at figure 1 it seems linear only in zone #2. Zone #1 and #3 have more convex shapes that could be better represented by quadratic equations (Poole, Colin F., and Sanka N. Atapattu. "Analysis of the solvent strength parameter (linear solvent strength model) for isocratic separations in reversed-phase liquid chromatography." Journal of Chromatography A 1675 (2022): 463153). Even though later in the text it seems that the predictions are good with the linear model.

Line 166, figure 2. I do not understand what do the legend in the graph means, what are those numbers? Is it the gradient steepness? It is not clear, not specified anywhere. And why some graphs have extrapolation for both shorter and longer gradients and other graphs only have one of the two?

Line 199-256 (selectivity section): This section could benefit from a series of practical (tangible) examples exemplifying the theoretical interpretations performed here. It is e.g. not clear how the selectivity in revHILIC (or in the other modes) would be affected if one where to change the column, mobile phase pH or additives. Hence how general are the drawn conclusions?

Line 283 (figure 4): it would help if the peptides would be numbered. Currently the comparison is unclear except that indeed overall the peak shapes are better in RPLC.

Line 304: Figure 5: What is Ce (%) exactly (specify here or in the supplementary info section – do not refer to literature for explanation). I find confusing compared to lines 213 stating that 1/(1+2.3b) represents retention (and hence not deltaCe)

Line 305, figure 5. Does not show RPLCxRPLC. Later, 2D RPLCxRPLC is also produced. So why not showing also the orthogonality for RPLCxRPLC?

Line 351: while complex, it might help to label at least some peaks in the contour plots. Otherwise the reader has not much to look at except for a number of undefined blobs in the contour plots.

Line 365 (and line 567): Why are such high temperatures used in the RPLC dimension? Isn’t this detrimental to the columns stability? Wouldn’t even better peak shapes be obtained at lower temperatures leading to even better transfer of the revHILIC plug?  

Line 377 (figure 7C): explain why the top left corner is an empty quarter of the contour plot. How cold this be solved

Line 418. It is claimed that UV data is not shown due to the chromatographic complexity, however, it would be interesting to see also UV chromatograms, eventually in a sample with reduced number of compounds, as it is the most used detector and the first choice for most HPLC analysis. Moreover, as they also suggest, it would make possible to calculate more realistic peak’s features (peak width, peak capacity…).

Line 452 (Figure 9): the time axes in (all) the second dimensions has been reversed. I find this very confusing.

Line 453-457, figures 8 and 9. Why no overlay and 3D plots also of RPLCxRPLC, as it has been done in previous figure?

line 498: what is the role of DDT?

Line 532: I believe the pressure release kit is not included or helpful to reduce the ripples in the (1D) detector but to protect the flow cell of the first dimension detector.

Typos:

-        Line 74 there in an extra parenthesis “)” at the end of the sentence

-        Line 223 “the the higher…” remove one “the”

Reviewer 3 Report

The article, molecules-2311065, “Reversed HILIC gradient: a powerful strategy for on-line comprehensive 2D-LC” treats interesting issue, and well presented. The reviewer considers that the article should be published after revisions as follows.

Comments

1.    Fig. 1 should contain error: the retention models (log k vs. % water) for 7 different peptides on the Waters BEH HILIC 133 column at 30°C, reached to log k = 2.5. That means k = 316, and it is too large if we consider the separation window is smaller than 25 min. The reviewer thinks that it should be ln k, instead of log k. The authors should confirm it.

2.    Throughout the manuscript, the terms, HILIC and revHILIC can be misunderstood as general meaning of HILIC system. The column, Waters BEH HILIC, seems to be a bare silica column, and cannot form thick water-enriched layer. It cannot be a representative column of many HILIC columns. So, the reviewer recommends to author to limit the meaning of the HILIC and revHILIC terms. The representative example is found in line 596-597.

3.    The use of a worse solvent for the sample preparation was discussed in lines 271 and 272. Why the author did not try to dissolve the sample in acetonitrile-water system, at least ? 

4.    In the paragraph that starts line 308, the authors discuss orthogonality based on the correlation factor R2, and it is interesting. Do authors have any theoretical background on the treatment, or any relevance to the results with those by the Stoll-Gilar bin-box method? Please add discussion on the method.

Followings are minor comments.

1.    Figure caption in Figure 2; in the graphs, capital letter is used, but in the caption, small letter is used to show A to F. It should be unified.

2.    Figure 2 are prepared with too small characters, and it is not easy to read.

3.    Equation 1 has different size font.

4.    Figure 5 are prepared with too small characters, and it is not easy to read.

Round 2

Reviewer 1 Report

This revised manuscript well addressed my previous comments. Overall, the clarity of the writing improved significantly, and the results are presented in a more accessible manner, thus reinforcing the conclusions drawn from the study. In the work, the authors assessed the efficacy of revHILIC for the analysis of small drugs and peptides and showed that it offers significant advantages over other techniques, including improved sensitivity and selectivity, when applied to 1D- and 2D-chromatographic analyses. I believe it will be a valuable contribution to the field of analytical strategies for biomolecules.